# Exploring Southern Ecuador’s Traditional Medicine: Biological Screening of Plant Extracts and Metabolites

**DOI:** 10.3390/plants13101422

**Published:** 2024-05-20

**Authors:** Nicole Bec, Christian Larroque, Chabaco Armijos

**Affiliations:** 1Institute for Regenerative Medicine and Biotherapy (IRMB), Université de Montpellier National Institute of Health and Medical Research (INSERM), 34295 Montpellier, France; nicole.bec@inserm.fr; 2Nephrology Department CHRU Montpellier, Institute for Regenerative Medicine and Biotherapy (IRMB), Université de Montpellier, 34295 Montpellier, France; cjlarroque@gmail.com; 3Departamento de Química, Universidad Técnica Particular de Loja, San Cayetano Alto, s/n. AP: 11 01 608, Loja 1101608, Ecuador

**Keywords:** traditional medicine of Ecuador, bioprospecting of natural products, cytotoxicity, cell-cycle analysis, subcellular targets

## Abstract

Ecuador stands as a nation inheriting a profound ancestral legacy in the utilization of medicinal plants, reflective of the rich biodiversity embraced by various ethnic groups. Despite this heritage, many of these therapeutic resources remain insufficiently explored concerning their toxicity and potential pharmacological effects. This study focused on a comprehensive evaluation of cytotoxicity and the potential subcellular targets within various extracts and nine isolated metabolites from carefully selected medicinal plants. Assessing their impact on the breast cancer cell line (MCF7), we subsequently examined the most active fractions for effects on the cell cycle, microtubule network, centrosome duplication, γH2AX foci, and E-cadherin. The investigated crude extracts and isolated compounds from Ecuadorian medicinal plants demonstrated cytotoxic effects, influencing diverse cellular pathways. These findings lend credence to the traditional uses of Ecuadorian medicinal plants, which have served diverse therapeutic purposes. Moreover, they beckon the exploration of the specific chemicals, whether in isolation or combination, responsible for these observed activities.

## 1. Introduction

According to the World Health Organization (WHO), more than 80% of the world’s population uses some form of traditional and complementary medicine at the primary level. Traditional and complementary medicine is defined by the WHO as comprising health practices, approaches, knowledge, and beliefs that incorporate plant-based medicines, spiritual therapies, manual techniques, and exercises. These are based on cultural traditions of healing that have been handed down from generation to generation and influenced by factors such as history, personal attitudes, and philosophy. Both traditional Chinese medicine as well as Ayurvedic practices maintained the knowledge and preserved the practice necessary for their survival [1].

For millions of people, herbal treatments, traditional remedies, and traditional medical practices represent the main, and sometimes only, source of health care. They are also culturally accepted. In “Western” civilization, the practice of traditional medicine was gradually lost and considered contrary to a rationalist application of patient care.

We can consider that the first obstacle to this dogma was brought by the description [2] then of the power [3] of the placebo effect and its necessary evaluation in the approval of any new drug.

The World Health Organization (WHO) now has a Traditional Medicine Strategy that supports the integration of these practices in national health systems [4]. In 2018, 98 countries, mostly from the African, South-east, South American, and Pacific regions but also including Canada, Germany, Switzerland, the UK, and Norway, had a national policy level for traditional medicine [4].

Natural products derived from plants play a role in the area of chemical medicine by providing a reservoir of secondary metabolites that have pharmacological actions. In the southern Andes of Ecuador, which has particularly rich biodiversity, a large variety of medicinal plants have been found to have biological activity as antioxidants and have anti-inflammatory, anti-parasitic, and many other pharmacological characteristics [5,6,7,8,9].

In the area of cancer, there is an urgent need for novel therapeutic agents to minimize the harmful side effects of cytotoxic chemotherapies, among which is the emergence of antibiotic-resistant bacteria that develop due to the weakened immune systems that render patients more sensitive to infections. In response to this pressing issue, a multifaceted approach is being pursued, encompassing diverse forms of therapy. This strategy involves a combination of therapies such as chemotherapy, immunotherapy with monoclonal antibodies [10,11] and immune system “boosters”, and radiation or complementary therapies. The synergy between these drugs is more potent than the efficacy of each one in isolation. These observations open the reflection on the utilization of complementary therapies based on the ancient recipes and traditional knowledge [12,13,14,15,16] compatible with approved therapeutic protocols. 

Based on this assumption, we took advantage of the extensive botanical biodiversity coupled with the traditional medicinal knowledge to screen various plant extracts for their cytotoxic potential.

For decades, products isolated from the plant kingdom such as taxol, vinca alkaloids, etoposides, and others have been established as reference molecules due to their notable anti-cancer efficacy [17]. Moreover, numerous studies have demonstrated the anti-tumoral potential of essentials oils [18] or various plant extracts [19,20].

The observed cytotoxicity of widely therapeutically used plant-derived drugs may affect diverse molecular pathways [21]. Cytotoxic compounds target, directly, the DNA of the cells and act as inhibitors of topoisomerase I (Camptothecin) or topoisomerase II [22]. The dynamics of microtubules are disrupted by molecules binding to tubulin, thus blocking the cell cycle [23]. Natural HDAC inhibitors that reactivate epigenetically silenced genes in cancer cells are promising candidates [19]. It is now admitted that the combination of drugs or multi-acting drugs for the simultaneous targeting of various pathways will be an option for new treatments [24]. These observations motivated us to investigate the impact of raw and complex extracts from plants used in traditional medicine on cell behavior, aiming to assess their potential in complementary treatments.

In this study, we focused on evaluating the potential targets of the selected plant extracts. To achieve this, we opted to investigate their influence on (1) the equilibrium of cell cycle phases, (2) microtubule network organization, (3) centrosome duplication, (4) DNA damage, and (5) E-Cadherin architecture. The selection of plants and their specific parts was based on comprehensive ethnobotanical, ethnopharmacological, and ethno-medical records developed in previous years [6,7,25,26,27].

## 2. Results

### 2.1. Screening the Extracts 

Plants were carefully chosen and collected based on local traditional medicinal practices and knowledge. When the traditional uses were clearly indicated, a brief description of these uses is provided in Table 1. A total of 71 plant extracts from two sampling campaigns involving 56 plant species were evaluated for their toxicity against the MCF7 breast cancer cell line. Concentrations of 20, 50, and 100 µg mL^−1^ for each extract, dissolved in DMSO, were selected arbitrarily for the initial screening. For the sake of clarity, only concentrations of 20 and 100 µg mL^−1^ are presented in the tables.

The outcomes of this evaluation are documented in the aforementioned tables. The selection of extracts for further experimentation was based on several factors. The initial criterion was the extracts’ ability to reduce the survival rate of MCF7 cell lines to less than 40% at a concentration of 50 µg/mL. An exception was made for the hydro-alcoholic extract from *Piper ecuadorense* Sodiro because of the wide use of this plant by local communities. Additionally, the solubility of the extracts in DMSO was taken into account. Sixteen extracts (numbered from one to sixteen) were then chosen and listed in Table 2.

### 2.2. Determination of the Cytotoxicity of the Selected Extracts

The survival of the MCF7 cell line was assessed following a 72 h exposure to concentrations ranging from 6.25 to 100 µg/mL of each extract. The IC_50_ was determined from this dataset as the concentration that prompts a 50% survival rate (see Section 4). These findings are outlined in Table 2. Notably, a majority of these plants boast recognized medicinal properties, which will be expounded upon below.

### 2.3. Description of Recognized Traditional Therapeutic Uses

*Cestrum sendtnerianum* C. Mart. (flowers) is used as a purgative and to treat head pain, stomach pain, fever, gangrene, influenza, internal infections, rheumatism, cough [4], postpartum relapse [25], and cold and to relieve inflammation in children after excessive exposure to sun [28].

*Croton lechleri* Müll. Arg. (latex) is used to treat hepatic pain and dermatitis; serves as a disinfectant; plays roles in the healing of wounds and diuretics [6]; acts against diarrhea, insect bites [5], gastritis, inflammation of the intestines, skin blemishes, and pimples; serves as an anti-parasitic and antiseptic; and is used to treat ulcers, throat infections, and gingivitis [29].

*Gaiadendron punctatum* (Ruiz & Pav.) G. Don (leaves) is used to treat bronchitis, hepatic pain, influenza, and cough; serves as a hair tonic [6]; fights strong cough [25]; and is also used as a hair tonic. It is used in case of measles and smallpox or for help in insomnia and to decrease disease relapses (*recaída* in Spanish) after childbirth [28].

*Garcinia macrophylla* Mart. (leaves and bark) is used for the relief of pains in the body and to treat external inflammation [25].

*Huperzia columnaris* B. Øllg. (aerial part): The fresh whole plant is used to treat liver and kidney diseases, fever, inflammation, and colds [30]. According to the main cultural tradition of the Saraguro people, this plant is mainly used as an intestinal purgative, especially to cure various supernatural diseases such as *vaho de agua* (exposure to water-vapors), *espanto, and susto* (startlement and fright) [31].

*Huperzia kuesteri* (Nessel) B. Øllg. (aerial part): The fresh whole plant treats liver and kidney diseases, fever, inflammations and colds [30]. According to Saraguro, people could take baths to relieve pain in the waist and backache, treat colds, give baths after childbirth, and use the plant as a purgative and to cure various supernatural diseases.

Hernandulcin: this [6-(1′-hydroxy-1′, 5 dimethyl-4′-hexenyl)-3-methyl-2-cyclohexenone] isolated product is 1000 times sweeter than sugar with any toxicity [33,34].

*Myrcianthes fragrans* (Sw.) McVaugh (leaves): It is used as an infusion to treat respiratory problems, the inflammation of the throat and gums, tonsillitis, and stomatitis, and is used to treat vaginal infections [35]. Besides medicinal uses, the leaves have also been employed as natural aromatic ingredients in the traditional Ecuadorian drink *colada morada*, which is drunk on the Day of the Dead (on 2 November each year).

*Phyla strigulosa* (M. Martens & Galeotti) Moldenke (leaves and flowers): This plant is used to treat stomachache [29], cramps, diarrhea in children, and intestinal infections and serves as a tonic.

Pinostrobin [(2S)-5-hydroxy-7-methoxy-2-phenyl-2,3-dihydrochromen-4-one]:

This substance was proposed as being nontoxic for the MCF7 cell line (IC_50_> 50 µM) but proposed as a topoisomerase 1 inhibitor highlighting the therapeutic potential of pinostrobin as an anti-proliferative agent [36].

*Piper ecuadorense* Sodiro (leaves): This plant helps fight hangover, acts as a disinfectant, and helps in the healing of wounds [6]. This plant is used in mythological cases as *mal aire* (bad air) [25].

*Piper pseudochurumayu* (Kunth) C. DC. (leaves): This plant is used to provide analgesic, diuretic, digestive, dermatological, anthelmintic, antirheumatic, and antidiarrheal effects and treat respiratory infections [9].

*Stereocaulon ramulosum* (Sw.) Raeusch. (aerial part) is used to treat external infections as its antibiotic activity has been reported [32].

### 2.4. Identification of Potential Target for the Selected Extracts

#### 2.4.1. Cell Cycle

We first studied the effect of selected extracts on the equilibrium of cell cycle phases. In Figure 1 and the Table 3 are presented the significance of the cell cycle phases and their ratio including the SubG1 upon incubation with 25 µg/mL of each extract.

#### 2.4.2. Microtubule

The microtubule network is part of the cellular cytoskeleton involved in the maintenance of the cell shape, mobility, and intracellular trafficking [37], and it is also the main component of the mitotic spindle. Any disturbance in the microtubule dynamics leads to cell cycle blockage and cell death [38]. Figure 2’s lane A illustrates a selected example of an observed effect on the microtubule network after the treatment using the extracts (here, in **13** and **3**). The effects of the 16 extracts and DMSO vehicle are presented in Appendix A.

#### 2.4.3. Centrosomes

Centrosomes are the microtubule-organizing centers (MTOCs) of eukaryotic cells [39]. The progress of mitosis depends on their duplication and their migration on either side of the nucleus during the G2 phase of the cell cycle.

Centrosome amplification or abnormality in their duplication lead to numerous diseases including cancer [40]. Upon the addition of the selected extracts, we evaluated the centrosomal integrity as follows. Typical features observed after incubation with compounds **1** or **4** are represented in Figure 2, lane B. The effects of the 16 extracts and DMSO vehicle are presented in Appendix A.

#### 2.4.4. DNA Damage Analysis

Maintaining DNA integrity is vital for cellular survival. However, DNA damage can occur depending on the cellular environment. Common types of damage include single- or double-strand breaks induced by irradiation, reactive oxygen species, or chemical agents. Additionally, cross-linking between strands and base damage caused by chemical modifications or adduct formation are frequent occurrences. In healthy cells, DNA repair mechanisms can often overcome such damage and restore DNA integrity [41].

DNA damage due to double-strand breakage is associated with the formation of γ-H2AX foci at the site of a break. This is one of the markers of the genotoxicity induced by a treatment or the environment of the cell. Genetic instability is one of the important factors that could drive cells to a tumoral phenotype [42]. Figure 2, lane C illustrates the detection of γ–H2AX foci observed in the cell nucleus after the treatment of the cell culture with extract **4** or **10**. The effects of the 16 extracts and DMSO vehicle are presented in Appendix A.

#### 2.4.5. E-Cadherin Test

E-Cadherin is a transmembrane protein regulating epithelial cell–cell adhesions that drives cellular proliferation and tissue morphogenesis. Thus, its expression is implicated in tumor progression and metastasis [43]. Its interaction with the F-actin network through the cadherin–catenin complex contributes also to dynamic cell movements in response to physical changes in the cells’ environment. When observed, the effect of the addition of plant extract to the culture medium was presented (Figure 2, lane D). The effects of the 16 extracts and DMSO vehicle are presented in Appendix A.

## 3. Discussion

All plants included in this study were chosen due to their utilization in the traditional medicine practices of southern Ecuador. The traditional uses of the plants from which extracts were prepared are described in the results section. Furthermore, certain extracts selected for the second phase of this study have been previously investigated for their pharmacological or medicinal properties, albeit without specifying their targets. For instance, ethanol extract from *Cestrum sendtenerianum* (**1**) was found to contain steroidal saponins exhibiting modest cytotoxic activity [44]. *Croton lechleri* (**2**), commonly known as Dragon’s blood, has been extensively studied, revealing diverse pharmacological benefits [5] including anti-tumoral effects [45], the management of diarrhea associated with AIDS or cancer treatments [16,46], and dermatological disease [47]. Cytotoxic compounds were extracted from *Garcinia macrophylla* (**4**) with potential anti tumoral properties [48]. An Acetyl cholinesterase inhibitory potential was demonstrated for preparations from *Huperzia columnaris* [31]. The essential oil of *Myrcianthes fragrans* (**8**) exhibited antimicrobial activity [34] and showed cytotoxic activity against the Hep G2 cell line [49]. Both the antifungal activity of raw extracts from *Piper ecuadorense* (**14**) [50,51] as well as larvicidal and antimalarial [52] activities of essential oil or extracts of *Piper pseudochurumayu* (**15**) have been demonstrated. Lastly, antimicrobial activity was found in an extract derived from *Stereocaulon* sp. (**16**) Ecuadorian lichen [53].

Following an extensive screening of extracts from plants traditionally employed in folk medicine and a subsequent selection of the most potent compounds, this study was conducted to enhance our understanding of some of these extracts by investigating their impact on potential cellular targets.

Table 3 illustrates the cell-cycle phase proportion after the culturing of MCF7 cells in the presence of 25 µg/mL of selected extracts. A well-described moderate increase in the G0/G1 phase due to DMSO [54] was observed. A 10% increase in the G0/G1 phase with a decrease in the S1 and G2/M phases was induced by compounds **4** and **5**, reflecting a possible blockage in the S phase entry. A strong increase in the proportion of cells in a Sub G1 phase, which are often referred to as apoptotic cells, is induced after incubation with the dilution of compounds **8**, **12**, **13**, and **15** without direct links with their overall toxicity (IC_50_) (Table 2). Then, incubation with 25 µg/mL of plant extract was investigated for its effect on cellular substructures, namely the microtubular network, centrosomes, DNA double-strand break, and E-cadherin assembly.

Among the extracts tested, extracts **3** (Figure 2, lane A, center) and **10** induced a disappearance of polymerized tubulin assemblies, but surprisingly, this effect was associated with strong toxicity only for extract **10**. Treatments with extracts **6**, **7**, **8**, **9**, **12**, **13** (Figure 2, lane A, right) and **16** resulted in a mis-organization of the microtubule with shorter filaments and a less dense network. Other extracts exerted no or slight effects on the microtubule network.

Concerning centrosomes that are the center for cellular microtubule organization, their behavior was investigated under the pression of the extracts (Figure 2, lane B). The average number of centrosomes visualized per cell calculated as described varied from 0.22 after treatment with extract **5** to 0.7 in the untreated condition. A significant decrease in the number of centrioles resulted from treatments with extracts **4** (Figure 2, lane B, right), **5**, **6**, and **14**, but this phenomenon was without correlation with toxicity or cell-cycle perturbation.

We evaluated the DNA breaks induced by the presence of the extracts in the cell culture medium through the visualization of the γ-H2AX foci into the cell nucleus. In Figure 2C, we evidence that the number of foci per cell nucleus was clearly increased after treatment with extracts **2**, **4** (Figure 2, lane C, center), **8**, and **14** and strongly increased after treatment with extract **10** (Figure 2, lane C, right).

Lastly, we analyzed the cell–cell adhesion parameter through the membrane expression of the E-cadherin protein. Upon vehicle-only addition, E-cadherin was, as expected, clearly expressed at the basal poles of the cells and concentrated at the cell–cell interface (Figure 2, lane D left). Extracts **6**, **10** and **15** clearly abolish the participation of E-cadherin in the membrane architecture whereas the others slightly modify its intracellular distribution.

Among the sixteen extracts tested, extracts **2**, **6**, and **10** exhibited the strongest cytotoxicity. For extract **10** from *H. columnaris,* the toxicity could be due to apoptosis following microtubule perturbation, the induction of double-strand breakage in the DNA, and the loss of cell–cell adhesion. For extract **6** from *H. kuesteri*, toxicity maybe related to centrosome duplication and apoptosis. In contrast, the analysis of the targets studied in this study did not allow us to explain the toxicity of extract **2** from *Croton lechleri*.

## 4. Materials and Methods

### 4.1. Chemical Compounds Studied in this Article

The following chemical compounds were included in our study: tricin (PubChem CID: 5281702); serratenediol (PubChem CID: 164947); 21-episerratenediol (PubChem CID: 12309682); serratenediol-3-O-acetate, pinostrobin (PubChem CID: 73201), pallidine (PubChem CID: 12313923), and O-methylpallidine (PubChem CID: 10405046); the flavone 5-hydroxy-4′,7-dimethoxyflavone (apigenin 7,4′-dimethyl ether) (PubChem CID: 5281601), and hernandulcin (PubChem CID: 125608).

### 4.2. Origin of Plant Material for Obtaining Extract Preparation

For the current study, various parts of medicinal plants, including leaves, flowers, roots, fruits, bark, latex, and aerial parts (leaves and stems), were collected between March 2009 and July 2017. This collection took place in the Loja and Zamora Chinchipe provinces, located in the southern region of Ecuador. These provinces are home to three significant traditional cultures of the South of Ecuador: *Campesinos*, *Shuar*, and *Saraguros,* as shown in Figure 3.

Plant identification was overseen by Bolivar Merino, Curator of the Herbarium Loja (HUNL) at the Universidad Nacional de Loja (UNL), who cross-referenced them with reference samples stored in the Herbarium. Voucher specimens of these plants were deposited at the Department of Chemistry of Universidad Técnica Particular de Loja (UTPL).

This collection process was carried out under the scientific investigation permission of the Ministry of Environment of Ecuador (MAE) under reference number No. 001-IC-FLO-DBAP-VS-DRLZCH-MA. Notably, the plant *Phyla strigulosa* was collected in March 2013 from the parish of Mejeche, canton Yantzaza, in province of Zamora Chinchipe, Ecuador. After collection, it was cultivated in a greenhouse at the conservation garden of Tumbaco (Pichincha province). Leaves were collected and selected by the Instituto Nacional de Investigaciones Agropecuarias (INIAP), and a botanical sample was prepared and assigned the voucher number MT-KN-111.

Ethnobotanical details, including the plant’s scientific name and family, common name in Spanish or *Kichwa* language, traditional medicinal use, laboratory extraction method, and the survival ratio of MCF7 cells in the presence of 20 or 100 µg/mL of each extract, are presented in Table 1 and Table 2. The systematic and nomenclature for each species were aligned with the Catalogue of the Vascular Plants of Ecuador [55] and the scientific names were cross-referenced with the database of http://www.theplantlist.org/ (accession date: 16 June 2017) [56].

### 4.3. Preparation of Extracts

All organic solvents used to extract the plants (EtOH, MeOH, MeCl_2_, EtOAc, and hexane) were reagent-grade and had been purchased from Sigma-Aldrich (St. Louis, MO, USA).

#### 4.3.1. Crude Extracts

Air-dried (35 °C) and milled plant parts selected (leaves, flowers, roots, fruits, bark, or latex) from each plant were separately extracted at one time at room temperature for two weeks with pure solvent reagent grade or a mixture of solvents (hexane, dichloromethane, ethyl acetate, methanol, ethanol, ethanol 70%, or ethanol 80%) as mentioned. Each filtrate was evaporated to dryness under reduced pressure at 40 °C to obtain 71 crude extracts that were kept in sealed amber glass vials at 4 °C until analysis.

#### 4.3.2. Lyophilized Extracts

The traditional aqueous preparation of guabiduca (*Piper crassinervium* Kunth), used as water of time, tonic, and diuretic, was purchased in a community assembly in the sector *Kiim*, in Zamora Chinchipe. The infusion was filtered, centrifuged, frozen at −40 °C, and lyophilized under vacuum in Labconco, model 7754047, series 10083033 (Kansas City, MI, USA) equipment for 72 h until dry powder was obtained. Dry extract was weighed, placed in a vial, labeled, and stored at −20 °C until its use.

#### 4.3.3. Alkaloid Extracts

Approximately 200–300 g of each selected dried plant (Table 1) was exhaustively extracted with a hydro-alcoholic mixture MeOH-H_2_O (80:20), then the solvent was removed via vacuum distillation, obtaining a dry extract (total extract), which was subjected to an acid-based extraction to obtain the alkaloid fraction [26, 31]. In this case, the total extract was suspended in dilute aqueous sulfuric acid (2% *v*/*v*), the suspension was filtered to separate the precipitated solids, and the process was repeated until negative reaction to the Dragendorff´s reagent. The combined aqueous phases were alkalinized until pH 11 by the addition of concentrated ammonia and extracted with chloroform until a negative reaction to the Dragendorff´s reagent occurred. Then, the combined organic phases were distilled under reduced pressure to remove the solvent and obtain the total alkaloid fraction of each plant selected for this study. The extracts of alkaloids were stored in dark flasks at 4 °C until they were analyzed.

#### 4.3.4. Pure Compounds

The pure compounds comprised three serratane triterpenoids, serratenediol (serrat-14-en-3*β*,21*α*-diol) (**1**), serratenediol-3-O-acetate (**2**), and 21-episerratenediol (serrat-14-en-3*β*,21*β*-diol) (**3**), that were isolated in a previous study from the aerial part of *Huperzia crassa* [31]. The flavone tricin (5, 7, 4′-trihydroxy-3′, 5′-dimethoxyflavone) (**4**) was isolated from the aerial part of *Huperzia brevifolia* as previously described [26,31]. The flavone pinostrobin (2s)-5-hydroxy-7-methoxyflavanone), compound (**5**), was isolated from the *Piper ecuadorense* species [50]. The alkaloids pallidine (**6**) and O-methylpallidine (**7**) were isolated from the aerial part of *Croton elegans* [57]. The flavone 5-hydroxy-4′,7-dimethoxyflavone (Apigenin 7,4′-dimethyl ether) (**8**) was isolated from *Piper peltatum* [58]. At the end, the compound hernandulcin ((6S)-6-[(2S)-2-hydroxy-6-methylhept-5-en-2-yl]-3-methylcyclohex-2-en-1-one), a sesquiterpene (**9**), was isolated from the species *Phyla strigulosa* [59].

The structures of the compounds (**1**–**9**) (see Figure 4) were identified on the basis of extensive spectroscopic analysis. The complete isolation process for each compound previously indicated is detailed in the reported bibliography.

The biological activity of the crude extracts obtained in Ecuador (Table 1) was evaluated in the Institut de Recherche enCancérologie de Montpellier, France. Two permits from the Ministry of the Environment of Ecuador (MAE) were used to send the extracts (crude and lyophilized) and pure compounds to France (MAE, No. 013-2014-IC-FAU-DPL-MA and MAE, No. 015-2014-IC-FAU-DPL-MA) under the authorization of scientific research project No. 047-IC-FLO-DPL-MA.

### 4.4. Biological Experiments

MCF7 HTB-22 cell line was from ATCC (Manassas, VA, USA), cell culture media and chemical reagents have been purchased from local distributors of Sigma-Aldrich (St. Louis, MO, USA).

#### 4.4.1. Cell Culture

All experiments were conducted on the MCF7 cell line, which was treated the day after plating in RPMI medium containing 10% FCS. Cells were exposed to the selected compounds for 72 h before being analyzed. DMSO was utilized for all treatments, with a final concentration of 1%. Each experiment was performed a minimum of twice.

#### 4.4.2. Cytotoxicity Evaluation

The viability of cells after concentration-dependent treatments was determined using the standard sulfo-rhodamine B assay, which measures cellular protein content. Cells were seeded at 5 × 10^3^ cells/well in 96-well plates. After treatment, cell monolayers were washed in phosphate buffer saline (PBS), fixed with 50% (*w*/*v*) trichloroacetic acid, and stained for 30 min in 0.4% sulfo-rhodamine B solution. The excess dye was then removed through washing repeatedly with 1% (*v*/*v*) acetic acid. The protein-bound dye was dissolved in 10 mM Tris base solution for OD determination at 540 nm using a microplate reader. IC_50_ values were determined from a nonlinear regression model using the online GNUPLOT package (www.ic50.tk, www.gnuplot.info, accessed on 30 April 2022).

#### 4.4.3. Cell-Cycle Analysis

The cell-cycle analysis was performed on MCF7 cell line seeded at 4 × 10^5^ cells/well in 6-well plate. After treatment, the cells were trypsinized, washed twice with chilled PBS, and spun in a cold centrifuge at 600 *g* for 5 min. The resulting pellet was fixed by resuspending it in 500 µL of chilled PBS followed by dropwise addition of 1.5 mL cold (−20 °C) ethanol. The cells were washed again twice with PBS, centrifuged at 600 *g* for 5 min, and resuspended in 500 µL PBS containing 100 µg/mL RNAse and 40 µg/mL propidium iodide. The staining reaction was allowed to proceed for 2 h at 37 °C. The DNA fluorescence was analyzed on a Gallios flow cytometer (Beckman Coulter France SAS, Villepinte, France). The results were analyzed using FlowJo cell-cycle analysis software (www.flowjo.com).

#### 4.4.4. Immunofluorescence Microscopy Analysis

Cells were seeded onto glass coverslips at 10^5^ cells/well in 6-well plates. Twenty-four hours later, cells were incubated for 72 h with 25 µg/mL of each extract excepted 12.5 µg/mL for extract **2**. After treatment, the cells were washed with PBS. For the visualization of microtubule network and centrosomes, the cells were fixed with cold (−20 °C) methanol for 10 min, gradually rehydrated with PBS, and incubated for 2 h at 37 °C with mouse anti-β-tubulin (clone TUB 2.1, T4026, Sigma, St. Louis, MO, USA) or rabbit anti-γ-tubulin (T3559, Sigma) primary antibodies diluted 1:500 in PBS-BSA 0.1%. For the visualization of γ-H2AX foci and E-cadherin, the cells were fixed with 3.7% paraformaldehyde plus 1% methanol for 5 min at 37 °C, permeabilized in PBS-Triton 0.1% for 5 min at 25 °C, washed in PBS, and then incubated for 1 h at 25 °C with mouse anti-phospho-histone H2A.X (Ser139) (clone JBW301, 05-636, Sigma) primary antibody diluted 1:1000 or mouse anti-cadherin (clone36E BD biosciences 610182) diluted 1/100 in PBS-Tween20 0.1% for 1 h at 37 °C. Subsequently, the coverslips were washed with PBS, incubated with anti-mouse or rabbit rhodamine-conjugated secondary antibodies (Rockland) diluted 1:200 in PBS-BSA 0.1% for 1 h at 37 °C. DAPI (4′, 6-diamidino-2-phenylindole) staining was then performed for 15 min followed by further PBS washing, air drying, and embedding in Mowiol-containing mounting medium. Fluorescence was detected using a Leica DM-RM microscope. Images were acquired with objective magnification 63× and processed using GIMP software version 2.10.34.

#### 4.4.5. Evaluation of the Effects on Centrosomes and γH2AX Foci

For this assessment, a minimum of three and up to six images were analyzed in a double-blinded manner for each condition. For centrosomes, the total number of centrosomes was divided by the cell number identified by their DAPI-stained nuclei. An average of 12 cells per image was counted for each analysis. The same methodology was applied for the counting of γH2AX foci.

## 5. Perspectives

In order to attain a more profound comprehension of the underlying mechanisms responsible for the tumoral cell-line toxicity observed in these plant extracts, it becomes imperative to undertake the tasks of isolating, characterizing, and evaluating pure compounds. Nevertheless, recent demonstrations of the “cocktail” effect, resulting from the synergistic potential of the combined presence of chemicals in concentrations that are individually inert [60,61], are a compelling impetus for us to intensify our efforts to study the complex formulations prepared by traditional healers from plant mixtures using the technical capabilities of modern medicine and biology [62,63,64,65].

## Figures and Tables

**Figure 1 plants-13-01422-f001:**
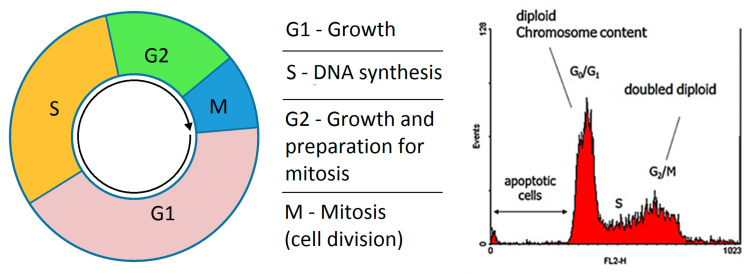
Description of cell cycle and illustration of a typical FACS experiment for the study of the extract effect on the cell cycle of MCF7 cell line incubated with each compound at 25 µg/mL for 72 h.

**Figure 2 plants-13-01422-f002:**
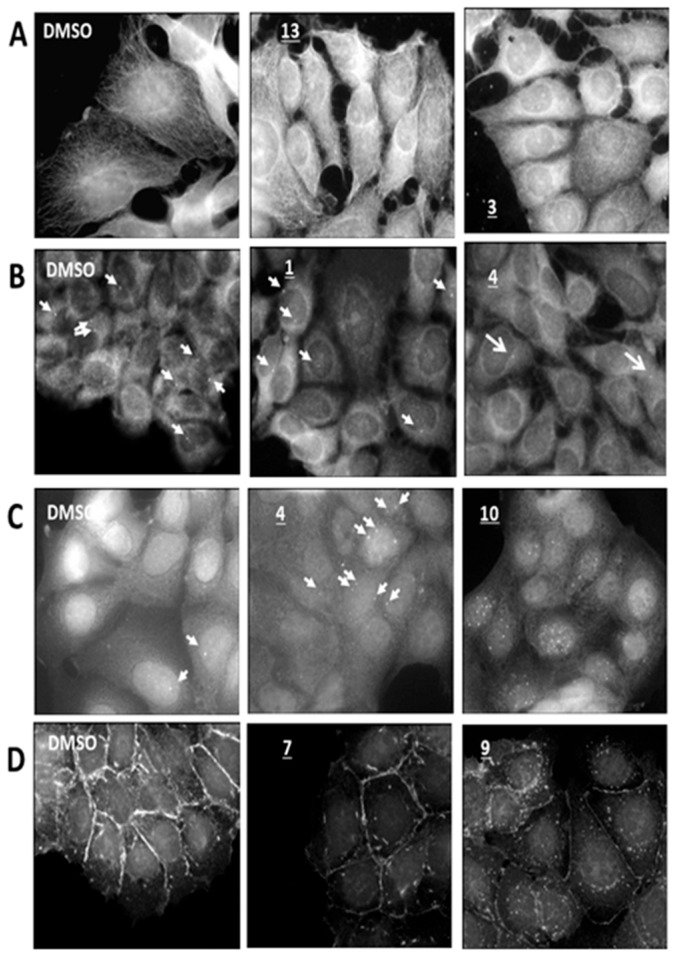
Illustration of the effects on cellular substructures observed after incubation of the MCF7 cell line in RPMI, 10% FCS at 37 °C, 5% CO_2_ supplemented with DMSO 1% or plant extracts at 25 µg/mL. Only two pictures for each cell structure are presented and are representative of the feature observed. For each lane, the following are shown: left pictures—reference cells cultured in the presence of 1% DMSO; center pictures illustrate a ‘moderate effect’ on the structure, and right pictures show the strongest observed effect. Label of the compound referring to its identity is inserted in each panel. Lane A: Microtubular network, Lane B: centrosomal substructure localized by white arrows, Lane C: gH2AX-positive DNA double-strand breaks localized with white arrows except in the right panel where they are to numerous. Lane D: E-Cadherin assembly. For clarity, merged picture with DAPI nuclear staining is not presented.

**Figure 3 plants-13-01422-f003:**
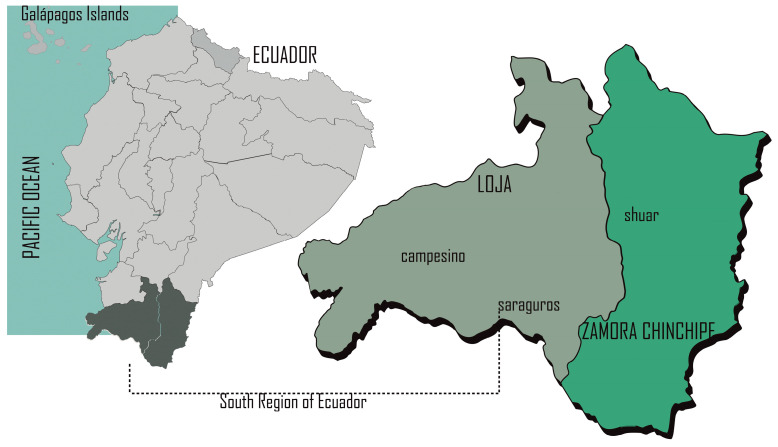
Settlement of healer communities from the south region of Ecuador, in Loja and Zamora Chinchipe cantons, that traditionally use plants that were of concern in this study.

**Figure 4 plants-13-01422-f004:**
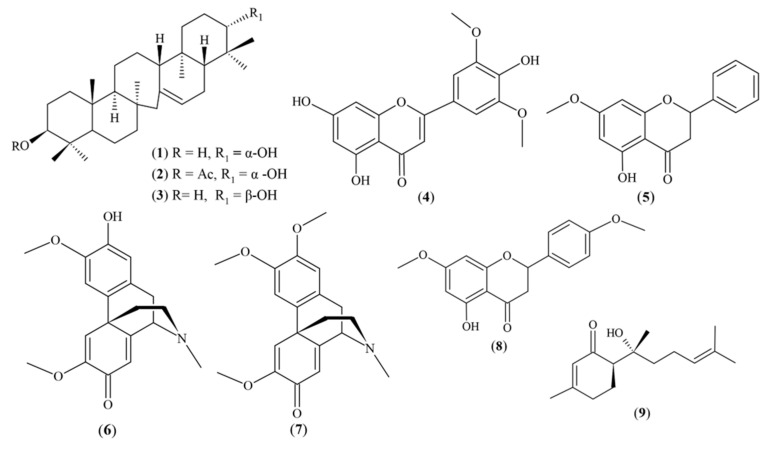
Pure compounds isolated from Ecuadorian plants.

**Table 1 plants-13-01422-t001:** Identification, traditional names, and usage of plants chosen and collected during the two campaigns. Plant parts used and secondary metabolite extraction conditions are presented. MCF 7 cell line survival after 72 h is reported in terms of percentage of surviving cells. Each plant extract identified as ‘selected compound’ is listed in Table 2 and subjected to further investigation to assess its impact on cellular pathways.

Plants and Compounds Evaluated	Family	Common Name	Traditional Uses	Extraction Condition	Product (µg/mL)	MCF Cell Line Survival Ratio (%)
**Control substances:**						
RPMI media	-	-	-	-	-	100
DMSO	-	-	-	-	-	100
TAXOL 1000 nM	-	-	-	-	0.8	18
TAXOL 10 nM	-	-	-	-	0.008	26
TAXOL 1nM	-	-	-	-	0.0008	94
**Pure compounds evaluated:**						
Serratenediol-3-O-acetate (C_32_H_52_0_3_)	-	Pure compound isolated from *Huperzia crassa* species		-	48	96
Tricin (5, 7, 4′-trihydroxy-3′, 5′-dimethoxyflavone) C_17_H_14_O_7_	-	Pure compound isolated (Flavone) from various species from *Huperzia* gender		PubChem CID: 5281702	33	88
5-hydroxy-4′,7-dimethoxyflavone (apigenin 7,4′-dimethyl ether) C_17_H_14_O_5_	-	Pure compound isolated from *Piper peltatum* L.		PubChem CID: 5281601	30	27
21-episerratenediol (serrat-14-en-3β,21β-diol) (C_30_H_50_O_2_)	-	Pure compound isolated from *Huperzia crassa*		PubChem CID: 12309682	100	47
					20	97
Serratenediol (serrat-14-en-3β,21α- diol) C_30_H_50_O_2_		Pure compound isolated from *Huperzia crassa*		PubChem CID: 164947	100	109
					20	100
Pinostrobin (2s)-5-hydroxy-7-methoxyflavanone) C_16_H_14_O_4_		Pure compound isolated from *Piper ecuadorense*		PubChem CID: 73201	100	60
					20	37
Pallidine (2-hydroxy-3,6-dimethoxy-17-methyl-5,6,8,14-tetradehydromorphinan-7-one) C_19_H_21_NO_4_		Pure compound isolated from *Croton elegans*		PubChem CID: 12313923	100	74
					20	103
O-methylpallidine ((1S,9S)-4,5,13-trimethoxy-17-methyl-17-azatetracyclo[7.5.3.01,10.02,7]heptadeca-2,4,6,10,13-pentaen-12-one) C_20_H_23_NO_4_		Pure compound isolated from *Croton elegans*		PubChem CID: 10405046	100	108
					20	107
Hernandulcin ((6S)-6-[(2S)-2-hydroxy-6-methylhept-5-en-2-yl]-3-methylcyclohex-2-en-1-one) C_15_H_24_O_2_		Pure compound isolated from *Phyla strigulosa*		PubChem CID: 125608	100	10
					20	67
**Species evaluated/Herbarium voucher**						
*Acanthoxanthium spinosum* (L.) Fourr./PPN-as-039	Asteraceae	Casamarucha, cardo de tres puntas	Treat conditions of the prostate and kidneys and urinary tract infection (oral testimony); anti-inflammatory and blood purifier [28]	EtOAc extract (leaves)	100	31
					20	103
				MeCl_2_ extract (leaves)	100	14
					20	93
*Baccharis obtusifolia* Kunth/PPN-as-014	Asteraceae	Chilca, chilca redonda, shadán	Antimycotic, cold, rheumatism [6]	MeOH extract (leaves)	100	84
					20	94
*Croton elegans* Kunth/HUTPL536	Euphorbiaceae	Mosquera	Anti-inflammatory; powerful purgative; treatment of rheumatism, neuralgia, and bronchitis [9]	Alkaloid fraction from MeOH-H_2_O extract (8:2) (leaves)	100	46
					20	80
*Cinchona officinalis* L./PPN-ry-002	Rubiaceae	Cascarilla, quina	Stomach pain; fever; malaria; antimycotic [6]	EtOH extract (bark)	100	90
					20	93
*Clusia alata* Triana & Planch./HUTPL5081	Clusiaceae	Duco	Gastritis [28]	MeOH extract (leaves)	100	18
					20	91
*Croton lechleri* Müll. Arg./PPN-eu-003	Euphorbiaceae	Sangre de drago	Selected compound	Latex	100	5
					20	4
*Renealmia alpinia* (Rottb.) Maas./HUTPL 11186	Zingiberaceae	Kumpía	The leaves are used to treat rheumatism; a blue pigment is obtained from the fruit [29]	Lyophilized aqueous fruit extract	100	6
					20	71
*Garcinia macrophylla* Mart./HUTPL3841	Clusiaceae	Shora	Selected compound	MeOH extract (leaves)	100	14
					20	53
*Huperzia brevifolia* (Grev. & Hook.) Holub/PPNIc-10	Lycopodiaceae	Waminga verde	Liver and kidney diseases, fever, inflammation, colds [30]	Alkaloid fraction from MeOH-H_2_O extract (8:2) (leaves)	100	4
					20	71
*Huperzia columnaris* B. Øllg./PPNIc-09	Lycopodiaceae	Waminga oso	Selected compound	Alkaloid fraction from MeOH-H_2_O extract (8:2) (leaves)	100	3
					20	42
*Huperzia compacta* (Hook.) Trevis./PPNIc-02	Lycopodiaceae	Waminga roja	Acts as a purgative and to treat supernatural diseases [31]	Alkaloid fraction from MeOH-H_2_O extract (8:2) (leaves)	100	90
					20	94
*Huperzia crassa* (Humb. & Bonpl. ex Willd.) Rothm./PPNIc-05	Lycopodiaceae	Waminga amarilla	To treat the itching of the body [29]	Alkaloid fraction from MeOH-H_2_O extract (8:2) (leaves)	100	82
					20	98
*Huperzia espinosana* B. Øllg/PPNIc-08	Lycopodiaceae	Waminga oso warmi	Liver and kidney diseases, fever, inflammation, colds [30]	Alkaloid fraction from MeOH-H_2_O extract (8:2) (leaves)	100	4
					20	78
*Huperzia kuesteri* (Nessel) B. Øllg./Ly-HK-001	Lycopodiaceae	Waminga verde grande	Selected compound	Alkaloid fraction from MeOH-H_2_O extract (8:2) (leaves)	100	4
					20	73
*Huperzia tetragona* (Hook. & Grev.) Trevis./PPNIc-04	Lycopodiaceae	Trencilla roja	Treatment of elephantiasis and leprosy and to treat supernatural diseases [31]	Alkaloid fraction from MeOH-H_2_O extract (8:2) (leaves)	100	81
					20	93
*Hypericum lancioides* Cuatrec./PP-hy-001	Hypericaceae	Bura bura	Antidepressant effects; antioxidant, antimicrobial, and antiviral properties	MeCl_2_ extract (aerial part)	100	23
					20	88
*Lycopodium complanatum* L./Ly-001-08	Lycopodiaceae	Gateador, trencilla	In bathrooms during the postpartum period for bone pain in children	Alkaloid fraction from MeOH-H_2_O extract (8:2) (leaves)	100	51
					20	76
*Loricaria thuyoides* (Lam.) Sch. Bip./PPN-as-044	Asteraceae	Ushkuchaki	Used in baths after childbirth to treat hip pain and cold and treat the *mal aire* (bad air) [20]	EtOAc extract (leaves and stem)	100	93
					20	100
*Ludwigia peruviana* (L.) H. Hara./PPN-on-003	Onagraceae	Mejorana	Hepatic pain, diuretic, kidney problems [6]	MeOH-H_2_O extract (9:1) (leaves and stem)	100	54
					20	97
*Macrocarpaea lenae* J. R. Grant/PPN-gn-003	Gentianaceae	Tabaco de cerro	Fever or cold caused by cold air or strong winds locally known as *mal aire* (bad air) [25]	Alkaloid fraction from MeOH-H_2_O extract (8:2) (flowers and leaves)	100	52
					20	93
*Sarcorhachis sydowii* Trel./Pi-003-010	Piperaceae	Intiwaska	The infusion of the leaves is drunk to treat stomach pain [29]	MeOH-H2O extract (9:1) (leaves and stem)	100	56
					20	96
				Lyophilized aqueous leaves and stem extract	100	31
					20	93
*Oreopanax andreanus* Marchal/PPN-ar-003	Araliaceae	Pumamaki	Disinfectant, healing of wounds, dermatitis [6]	MeOH extract (leaves)	100	102
					20	97
*Curcuma longa* L./HUTPL 14333	Zingiberaceae	Cúrcuma, urmeric, perenchi	The plant is traditionally known for its fungicidal and bactericidal properties [9]	Lyophilized aqueous tuber extract	100	87
					20	85
*Piper pseudochurumayu* (Kunth) C. DC./PPN-pi-009	Piperaceae	Matico, ámbarámbar	Selected compound	MeOH extract (leaves and stem)	100	5
					20	54
*Siparuna eggersii* Hieron./PPN-mn-001	Monimiaceae	Monte del oso	Strokes, diabetes, fractured bones, rheumatism, kidney problems [6]	MeOH extract (leaves and stem)	100	39
					20	95
*Piper crassinervium* Kunth/PPN-pi-002	Piperaceae	Guabiduca	Diabetes, gastritis, prostate problems [6]	Lyophilized aqueous leaf extract	100	96
					20	106
*Juglans neotropica* Diels/PPN-ju-001	Juglandaceae	Nogal	Rheumatism, hepatic pain in postpartum bath [25]	Lyophilized aqueous leaf extract	100	5
					20	106
*Tropaeolum tuberosum* Ruiz & Pav./PPN-tr-001	Tropaeolaceae	Mashua	Prostate [25]	Lyophilized aqueous tuber juice	100	103
					20	105
*Valeriana pyramidalis* Kunth./FT991	Valerianaceae	Valeriana	To treat nerves, heart, liver, and kidney problems [29]	Lyophilized roots exudate	100	80
					20	105
*Piper ecuadorense* Sodiro/PPN-pi-007	Piperaceae	Matico grande, tiklilin grande, matico del monte	Selected compound	EtOH-H_2_O extract (7:3) (leaves)	100	18
					20	54
			Selected compound	MeOH extract (leaves)	100	24
					20	100
*Alibertia* sp.	Rubiaceae	Matiri	The fruits of several *Alibertia* species are edible [29]	MeOH extract (leaves)	100	43
					20	85
*Artemisia sodiroi* Hieron./PPN-as-021	Asteraceae	Ajenjo	Internal inflammation, stomach pain, hepatic pain, fever, internal infections, kidney problems, cough [6]	MeOH extract (leaves)	100	71
					20	111
*Artocarpus altilis* (Parkinson) Fosberg/PPN-mo-003	Moraceae	Fruto del pan	Diabetes, high cholesterol [6]	MeOH extract (leaves)	100	90
					20	104
*Bejaria resinosa* Mutis ex L.f./PPN-er-002	Ericaceae	Payama, pena pena, pena de cerro	To treat nervous system problems, swollen wounds and inflammations of the genital organs, as well liver diseases and cancer [8]	MeOH extract (leaves)	100	6
					20	89
*Brugmansia suaveolens* (Willd.) Bercht. & J. Presl/PPNso-021	Solanaceae	Floripondio rosado, guando rosado	To treat rheumatic pain [27]	EtOH-H_2_O (8:2) (flowers)	100	94
					20	110
*Brugmansia versicolor* Lagerh./PPN-so-027	Solanaceae	Floripondio, guando	To treat headache and inflammation and swelling from blows and act as psychoactive plant [29]	Alkaloid fraction from MeOH-H_2_O extract (8:2) (flowers)	100	84
					20	125
*Centropogon comosus* Gleason/HUTPL 11342	Campanulaceae	Motepela	Wash insect bites (oral testimony)	EtOH-H_2_O (7:3) (leaves)	100	94
					20	107
*Cestrum sendtnerianum* C. Mart./PPN-so-003	Solanaceae	Sauco negro	Selected compound	EtOH-H_2_O (7:3) (flowers)	100	6
					20	47
			Purgative, head pain, stomach pain, fever, gangrene, influenza, internal infections, rheumatism, cough [6]	MeOH extract (leaves and flowers)	100	149
					20	115
*Clusia alata* Triana & Planch/HUTPL5081	Clusiaceae	Duco	Gastritis [28]	MeOH extract (fruits)	100	9
					20	85
*Gallesia integrifolia* (Spreng.) Harms/PPN-ph-001	Phytolaccaceae	Palo de ajo	Arthritis, strokes, rheumatism [6]	MeOH extract (bark)	100	7
					20	88
				MeOH extract (leaves)	100	106
					20	107
*Gaiadendron punctatum* (Ruiz & Pav.) G. Don/PPN-lo-001	Loranthaceae	Violeta de cerro, violeta de campo	Strong cough [25]	EtOH extract (flowers)	100	28
					20	73
			Selected compound	EtOH extract (leaves)	100	5
					20	22
*Gaultheria erecta* Vent/PPN-er-008	Ericaceae	Mote pela	The fruits are edible [25]	EtOH-H_2_O (7:3) (flowers)	100	14
					20	78
*Huperzia weberbaueri* (Hieron. & Herter ex Nessel) Holub/PPNIc-07	Lycopodiaceae	Waminga suca	Purgative and to treat supernatural diseases [31]	Hexane extract (aerial part)	100	75
					20	93
				MeOH extract (aerial part)	100	58
					20	100
*Hesperomeles ferruginea* (Pers.) Benth./HUTPL4010	Rosaceae	Quique	The fruits can be used as foods [29]	EtOH-H_2_O (7:3) (fruits)	100	73
					20	100
				EtOH-H_2_O (7:3) (leaves)	100	17
					20	100
*Ilex guayusa* Loes./PPN-aq-001	Aquifoliaceae	Guayusa	Gastritis, relaxant, increasing woman’s fertility [6]	EtOH-H_2_O (7:3) (leaves)	100	5
					20	92
*Iresine herbstii* Hook./PPN-am-001	Amaranthaceae	Escancel	Fever, relaxant, kidney [6]	Lyophilized aqueous (leaves and stems)	100	8
					20	78
				EtOH-H_2_O (7:3) (leaves and stems)	100	5
					20	93
*Lupinus semperflorens* Hartw. ex Benth./HUTPL4786	Fabaceae	Chocho silvestre, taure de cerro, aspa chocho	Fever and stomach pain	MeOH extract (leaves and stems)	100	56
					20	127
*Salvia pichinchensis* Benth/PPN-la-014	Lamiaceae	Matico negro, matico grande de cerro	To treat the infection of external wounds and for curing kidney and liver disorders [9]	EtOH-H_2_O (7:3) (leaves and stems)	100	9
					20	122
*Myrcianthes fragrans* (Sw.) McVaugh/PPN-my-008	Myrtaceae	Arrayán aromático, saco, wawall (*kichwa*)	Selected compound	MeOH extract (leaves)	100	6
					20	63
*Oreopanax ecuadorensis* Seem./PPN-ar-001	Araliaceae	Pumamaqui	Headache [6]	MeOH extract (leaves)	100	87
					20	100
*Oreopanax eriocephalus* Harms/HUTPL 4901	Araliaceae	Maqui-maqui	Anti-inflammatory and antibacterial properties [9]	MeOH extract (leaves and flowers)	100	87
					20	96
*Otholobium mexicanum* (L. f.) J.W. Grimes/PPN-fa-005	Fabaceae	Culén, teculén	Stomach pain, diarrhea, indigestions, contraceptive [6]	EtOAc extract (leaves and flowers)	100	4
					20	93
*Phyla strigulosa* (M. Martens & Galeotti) Moldenke/MT-KN-111	Verbenaceae	Buscapina, novalgina	Selected compound	EtOAc extract	100	6
					20	61
			Selected compound	Hexane extract (leaves and flowers)	100	7
					20	15
			Stomachache [29], cramps, diarrhea in children, and intestinal infections; to act as tonic	Lyophilized aqueous leaves and flowers extract	100	13
					20	102
			Selected compound	MeOH extract (leaves and flowers)	100	6
					20	38
*Cestrum racemosum* Ruiz & Pav./PPN-so-010	Solanaceae	Sauco, sauco de montaña, sauco blanco	Tooth decay, headache, stomach pain, fever, gastritis [6]	MeOH extract (leaves and stem)	100	24
					20	91
*Stereocaulon ramulosum* (Sw.) Raeusch./MUTPL-AB-0650	Stereocaulaceae	Musgo	External infections, antibiotic [32]	EtOAc extract (aerial part)	100	7
					20	71
			Selected compound	MeCl_2_ (aerial part)	100	7
					20	61
*Echinopsis pachanoi* (Britton & Rose) Friedrich & G.D. Rowley/PPN-cb-001	Cactaceae	San Pedro cactus with 5 ribs/San pedrillo	To induce visions (oral and inhaled administration), to act as a purgative, to treat supernatural diseases, to treat anxiety, and serve as an anti-inflammatory or wound disinfectant [27]	Lyophilized from the aqueous extract	100	41
					20	94
		San Pedro cactus with 7 ribs/San pedrillo		Lyophilized from the aqueous extract	100	7
					20	68
		San Pedro cactus with 9 ribs/San pedrillo		Lyophilized from the aqueous extract	100	19
					20	66

**Table 2 plants-13-01422-t002:** IC_50_ of the extracts selected from the screening. The toxicity of each compound is measured as described in the Materials and Methods section. IC_50_ is defined as the concentration deduced from these data as the concentration that leaves 50% of cells alive. Identification, family, common name of the plant, and extraction mode are also reported.

New Label	Plant Identity	Common Name	Extraction Mode	IC_50_ vs. MCF7 Cell Line (µg/mL)	Error Bar (µg/mL)	Error Bar (%)
1	*Cestrum sendtnerianum* C. Mart.	Sauco negro	EtOH-H_2_O (70:30)	36.80	3.62	9.82
2	*Croton lechleri* Müll. Arg.	Sangre de drago	Latex	5.63	0.00	0.00
3	*Gaiadendron punctatum* (Ruiz & Pav.) G. Don	Violeta de campo, violeta de cerro	EtOH	15.62	0.20	1.28
4	*Garcinia macrophylla* Mart.	Shora	MeOH	36.72	1.20	3.28
5	*Huperzia columnaris* B. Øllg.	Waminga oso	Alkaloid fraction	27.35	1.81	6.62
6	*Huperzia kuesteri* (Nessel) B. Øllg.	Waminga verde grande	EtOAc	5.39	3.23	59.80
7	Hernandulcin ((6S)-6-[(2S)-2-hydroxy-6-methylhept-5-en-2-yl]-3-methylcyclohex-2-en-1-one)	-	Pub Chem CID: 125608	29.95	4.64	15.50
8	*Myrcianthes fragrans* (Sw.) McVaugh	Arrayán aromático, saco, wawall	MeOH	36.02	5.62	15.60
9	*Phyla strigulosa* (M. Martens & Galeotti) Moldenke	Novalgina, buscapina	EtOAc	39.53	1.03	2.59
10			Hexane	10.27	0.23	2.21
11			MeOH	28.83	5.66	19.64
12	Pinostrobin ((2s)-5-hydroxy-7-methoxyflavone)	-	Pub Chem CID:73201	23.85	3.82	16.01
13	*Piper ecuadorense* Sodiro	Matico grande, tiklilin grande, matico del monte	EtOH-H_2_O (70:30)	79.82	2.32	2.90
14			MeOH	30.69	21.88	71.28
15	*Piper pseudochurumayu* (Kunth) C. DC.	Matico, ámbar ámbar	MeOH	21.39	1.14	5.31
16	*Stereocaulon ramulosum* (Sw.) Raeusch.	Musgo	MeCl_2_	19.26	3.26	16.91

**Table 3 plants-13-01422-t003:** Proportion of each cell cycle phase of the MCF7 cell line treated for 72 h with 25 µg/mL of each extract dissolved in RPMI, 10% FCS.

Extract From:	Label	G0/G1	S	G2/M	Apoptosis
	RPMI	63.0	16.2	20.9	0.6
	DMSO (1%)	76.4	10.7	11.5	1.8
*Cestrum sendtnerianum* (flower)	1	74.4	12.1	12.5	1.7
*Croton lechleri*	2	73.0	12.9	13.4	1.5
*G. punctatum* (levaes)	3	72.8	12.4	14.3	0.9
*Garcinia macrophylla*	4	82.6	7.9	5.9	3.8
*H. columnaris*	5	83.2	7.8	7.4	2.0
*H. kuesteri*	6	64.4	15.3	9.8	12.7
Hernandulcin	7	75.5	11.9	6.2	9.4
*Myrcianthes fragrans*	8	66.9	16.3	7.5	12.6
*Phyla strigulosa*	9	73.0	13.1	8.2	8.1
*Phyla strigulosa*	10	70.0	14.4	10.7	6.8
*Phyla strigulosa*	11	70.6	15.4	8.8	8.1
Pinostrobin	12	69.8	14.5	6.4	11.8
*Piper ecuadorense*	13	69.0	15.0	7.5	12.0
*Piper ecuadorense*	14	72.4	13.8	7.8	8.6
*Piper pseudochurumayu*	15	69.7	14.2	9.5	9.1
*Stereocaulon ramulosum*	16	72.5	13.2	9.1	7.4

## Data Availability

Data will be made available on request.

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
