# Peer review of "Exploring Southern Ecuador’s Traditional Medicine: Biological Screening of Plant Extracts and Metabolites"

_plants, 2024, doi:10.3390/plants13101422_

Round 1

Reviewer 1 Report

Comments and Suggestions for Authors

Well written, interesting contribution to the better understanding of medicinal plant resources of Southern Equador.

As a comment, Reviewer would like to mention that it would be worth highlighting the originality of these findings, as compared to the available - most probably - not too rich reference sources.

Comments on the Quality of English Language

Well written manuscript. No comments.

Author Response

REVIEWER #1.

Well written, interesting contribution to the better understanding of medicinal plant resources of Southern Equador.

-As a comment, Reviewer would like to mention that it would be worth highlighting the originality of these findings, as compared to the available - most probably - not too rich reference sources.

We thank the referee for encouraging comment and hope that this work will be useful to those who work to recognize the potential of traditional medicines.

Reviewer 2 Report

Comments and Suggestions for Authors

Line 75. Add another reference (Beside Ref 15), because there is a large number of published works with evidence of antimutagenic and anticarcinogenic activity of essential oils, plant extracts as well as individual components. List some other herbal compounds with this activity.

Line76-83. List some other mechanisms of DNA repair for which there is ample evidence that plants can induce and thereby enable DNA repair. Review the literature data well, because the reparation of induced DNA damage by plants and their compounds has been done for a long time.

In Table 3, it is not clear what the first column means

Line 362 : In 2.4.4. Indicate that this is only one type of DNA damage out of many that can occur. Name some of them. List what all can induce DNA damage, at least some of them since you already mention the environment.

Line 271: specify which test was used

Line 419. The first and second sentences are unclear, unfinished.

Author Response

ANSWERS to REVIEWERS

We have addressed all the issues raised by the reviewers who are thanked for having improved the quality of the manuscript. Here are our answers, point by point.

REVIEWER #2. Changes have been highlighted in yellow.

-Line 75. Add another reference (Beside Ref 15), because there is a large number of published works with evidence of antimutagenic and anticarcinogenic activity of essential oils, plant extracts as well as individual components. List some other herbal compounds with this activity.

Of course, we agree with the referee for this comment. The references of recent studies and review are now added.

-Line76-83. List some other mechanisms of DNA repair for which there is ample evidence that plants can induce and thereby enable DNA repair. Review the literature data well, because the reparation of induced DNA damage by plants and their compounds has been done for a long time.

We agree with the referee's observation that numerous DNA interacting plant-derived products have been extensively documented over time. However, our focus in this paragraph is specifically on citing anti-cancer drugs derived from plants that are widely acknowledged and utilized in established clinical protocols.

-In Table 3, it is not clear what the first column means

For more clarity tables 1 and 2 were combined and the table 3 (now table 2) was completed.

-Line 362: In 2.4.4. Indicate that this is only one type of DNA damage out of many that can occur. Name some of them. List what all can induce DNA damage, at least some of them since you already mention the environment.

We agree with the referee that many type of DNA damage could occur. A short paragraph describing some DNA damage type and the formation of gH2Ax foci was added.

-Line 271: specify which test was used

All tests used are described under materials and method.

-Line 419. The first and second sentences are unclear, unfinished

The text was modified.

Reviewer 3 Report

Comments and Suggestions for Authors

The manuscript ˝Exploring Southern Ecuador's Traditional Medicine: Biological Screening of Plants Extracts and Metabolites˝ is an interesting scientific manuscript that provides data on numerous plant extracts. However, the manuscript is not well written, the relevant data on extract activity are missing, and the tables as well as Figures are not clearly represented.

Please find my suggestions as follows:

Lines 31-43 - a proper reference is missing.

Lines 48 and 52 – WHO should be added to the reference list

The Introduction part is not very well written. The authors should explain the mechanisms of cancer that involve selected targets tested in the manuscript. For example, how is the section between lines 59 and 68 relevant to this research? The authors did not test combinations of herbal drugs.

As the authors stated in section 4.3.1, the extracts were prepared using different solvents. Can you explain why did you chose different solvents for different plant species?

Tables 1 and 2 have too much information and are not transparent. For example, there is no need to include family and common name. Referent compounds should be expressed also in µg/ml.

Why are pure compounds from Table 1 tested in different concentrations than compounds from Table 2 and on what basis did you select these specific compounds?

Is there a specific reason for including undefined aqueous infusions in this research?

I would recommend making a new table with relevant data, namely plant name, and name of the organ used for extraction (leaves, fruit, flower…).  There is also no point in presenting the MCF7 cell line survival ratio for two concentrations. If the 16 selected extracts for further research were selected based on their activity at the concentrations of 50 µg/mL, then this is the only relevant concentration that should be in the tables.

Please provide a graphical representation with IC50 values (instead of Table 3) for the tested plants and include standard deviations.  

If the authors omit the ˝traditional uses˝ in  Tables 1 and 2, then section 2.3 is relevant to the manuscript. If the authors insist on keeping the ˝traditional uses˝ in the tables then section 2.3. should be left out since it provides the same data.

Part 2.4 needs to be described in more detail. Please provide an explanation of potential targets and how the impact on every target reflects on cancer growth.

Figure 2. – please provide photographs of all tested extracts. For clarity, you can divide it into several Figures.

The Discussion part should be better elaborated. The first part of the discussion provides insights into previous cytotoxicity research, but the obtained results must be better explained (mechanism of action, activity, etc.) and supported by relevant literature (for example, some examples of plant extracts with similar activity in tested biological assays, comparison of their cytotoxicity)

Line 480 – the Plantlist webpage should be listed in the reference part and enumerated as a reference.

Line 482/483 – this sentence is more appropriate for the Acknowledge or Funding part.

Part 4.3. – the exact solvents used should be named.

Line 535-537 – please specify the bibliography for the isolation process.

Part 4.4. – please provide relevant references for the assay protocol

The conclusion is not relevant to this manuscript.

The page numbers are not in order.

The references are outdated, please include more recent papers.

Comments on the Quality of English Language

Moderate editing of English language required

Author Response

ANSWERS to REVIEWERS

We have addressed all the issues raised by the reviewers who are thanked for having improved the quality of the manuscript. Here are our answers, point by point.

REVIEWER #3. Changes have been highlighted in light blue.

-The manuscript ˝Exploring Southern Ecuador's Traditional Medicine: Biological Screening of Plants Extracts and Metabolites˝ is an interesting scientific manuscript that provides data on numerous plant extracts. However, the manuscript is not well written, the relevant data on extract activity are missing, and the tables as well as Figures are not clearly represented.

We hope that all modifications will satisfy the referee.

Please find my suggestions as follows:

-Lines 31-43 - a proper reference is missing.

A general WHO reference addressing this point is added.

-Lines 48 and 52 – WHO should be added to the reference list

Done

-The Introduction part is not very well written. The authors should explain the mechanisms of cancer that involve selected targets tested in the manuscript. For example, how is the section between lines 59 and 68 relevant to this research? The authors did not test combinations of herbal drugs.

A sentence was added to clarify this point: “These observations motivated us to investigate the impact of raw and complex extracts from plants used in traditional medicine on cell behavior, aiming to assess their potential as complementary treatments.”

-As the authors stated in section 4.3.1, the extracts were prepared using different solvents. Can you explain why did you chose different solvents for different plant species?

Each solvent has an affinity for different types of compounds. Some solvents may be more efficient than others in the extraction of certain compounds, and therefore their biological activity may be different.

-Tables 1 and 2 have too much information and are not transparent. For example, there is no need to include family and common name. Referent compounds should be expressed also in µg/ml.

For more clarity tables 1 and 2 were modified and combined in the new table 1.

-Why are pure compounds from Table 1 tested in different concentrations than compounds from Table 2 and on what basis did you select these specific compounds?

In the present study, an attempt was made to evaluate the cytotoxicity of several extracts of plants used in traditional medicine in Ecuador, but since in some of the previous studies with certain plants were obtained considerable quantities of pure compounds, it was decided to evaluate them and provide data on their toxicity that allow us to better understand their therapeutic action. Regarding the substances we use to evaluate, not all of them are available in significant quantities.

-Is there a specific reason for including undefined aqueous infusions in this research?

We haven’t no more information on the plants used for these preparations, so we delete corresponding lines from the table 1.

-I would recommend making a new table with relevant data, namely plant name, and name of the organ used for extraction (leaves, fruit, flower…). 

Done

-There is also no point in presenting the MCF7 cell line survival ratio for two concentrations. If the 16 selected extracts for further research were selected based on their activity at the concentrations of 50 µg/mL, then this is the only relevant concentration that should be in the tables.

We have opted to provide the survival ratios at concentrations of 20 and 100 µg/ml for each of the 80 extracts tested. This allows the reader to gain insight into the toxicity range of all extracts.

-Please provide a graphical representation with IC50 values (instead of Table 3) for the tested plants and include standard deviations.  

Table 2 was modified and completed. We don’t think that adding a graphical representation, as follow, would help the reader.

-If the authors omit the ˝traditional uses˝ in  Tables 1 and 2, then section 2.3 is relevant to the manuscript. If the authors insist on keeping the ˝traditional uses˝ in the tables then section 2.3. should be left out since it provides the same data.

The traditional uses described in section 2.3 refer solely to the 16 compounds selected for studies on cellular culture. Their traditional uses are not described in table 1. These compounds are referred to as 'selected compound' in the new table 1.

-Part 2.4 needs to be described in more detail. Please provide an explanation of potential targets and how the impact on every target reflects on cancer growth.

Short description and references are added to illustrate the implication in cancer development of each biological process targeted.

-Figure 2. – please provide photographs of all tested extracts. For clarity, you can divide it into several Figures.

Four compilations of photographs for all tested extracts are now included; but for more readability of the paper we propose to keep the figure 3 as it and include the set of four panels as supplementary material: Figure 2S-1: Microtubules network upon incubation with extracts; Figure 2S-2: Centrosomal gTub localization upon incubation with the extracts; Figure 2S-3: Nuclear gH2aX foci formation upon incubation with the extracts ; Figure 2S-4 : Cadherine localization upon incubation with extracts.                      

-The Discussion part should be better elaborated. The first part of the discussion provides insights into previous cytotoxicity research, but the obtained results must be better explained (mechanism of action, activity, etc.) and supported by relevant literature (for example, some examples of plant extracts with similar activity in tested biological assays, comparison of their cytotoxicity).

The mechanisms of action of the extracts on each pathway explored are complex and not necessarily directly linked to cytotoxicity. This study should be regarded as a screening of extracts from plants with traditional medicinal use in Southern Ecuador. We believe that any mechanistic considerations regarding each pathway tested require specific studies to avoid being overly speculative. Some experiments aimed at addressing this are currently underway.

-Line 480 – the Plantlist webpage should be listed in the reference part and enumerated as a reference.

A new reference has been added: Plantlist.org. 2013. http://www.theplantlist.org (accessed on 16 June 2017)

-Line 482/483 – this sentence is more appropriate for the Acknowledge or Funding part.

This sentence has been removed.

-Part 4.3. – the exact solvents used should be named.

In this section the name of all the solvents has been added: All organic solvents used to extract the plants (EtOH, MeOH, MeCl 2 , EtOAc, and hexane) were reagent grade and were purchased from Sigma-Aldrich.

-Line 535-537 – please specify the bibliography for the isolation process.

We have added a new bibliography: “Vega M, Brito B, Malagón O. 2017. Application of qNMR in the characterization of hernandulcin in the species Phyla strigulosa. Conference Paper. XXVI Congresso SILAE. At: Cartagena de Indias, Colombia.”

-Part 4.4. – please provide relevant references for the assay protocol.

Cellular culture condition and survival evaluation were described, and the software used for IC50 determination is now cited.

-The conclusion is not relevant to this manuscript.

We agree with this comment and change “Conclusion” for “Perspectives”.

-The page numbers are not in order.

 Corrected

-The references are outdated, please include more recent papers.

Done

Round 2

Reviewer 3 Report

Comments and Suggestions for Authors

Table 1. – concentrations of Taxol  - please correct concentration  in µg/ml of Taxol 1000 nM (it cannot be the same as concentration of Taxol 1 nM)

Table 1 – please check carefully all compounds listed in the Table 1.  Correct the names for the first two compounds and check CIDs. For example, 5-hydroxi-4,7-dimetoxi-fla-vone is not equivalent to PubChem CID: 12309682

Table 1 – please explain See selected compounds (it has been written several times in the Table 1 ., it would be useful to explain in the Tables´ title what it means, or put an asterisk and explain it under the table)

Table 1 – please include traditional use for Phyla strigulosa

You have stated that the initial criterion for further experiments (results showed in Table 2) was the extracts' ability to reduce the survival rate of MCF7 cell lines to less than 40%  (so the rate of cell survival was from 0-40%) at a concentration of 50 μg/ml. How can you than have IC50=79.82 µg/mL for Piper ecuadorense Sodiro?

Please check the cited literature as it is not always listed in the same citation style.

Comments on the Quality of English Language

Minor editing reqiured

Author Response

Answer to reviewers: Round 2

We have addressed all the issues raised by the reviewer who are thanked for having improved the quality of the manuscript. Here are our answers, point by point. In the latest version of the manuscript the suggested changes and other made by us are highlighted in green.

REVIEWER #3. Changes have been highlighted in green.
-Table 1. – concentrations of Taxol - please correct concentration in μg/ml of Taxol 1000

nM (it cannot be the same as concentration of Taxol 1 nM)

We thank the referee for mentioning our mistake (ng/ml was omitted). For homogeneity all concentrations are now in μg/ml.

-Table 1 – please check carefully all compounds listed in the Table 1. Correct the names for the first two compounds and check CIDs. For example, 5-hydroxi-4,7-dimetoxi-fla-vone is not equivalent to PubChem CID: 12309682

Some compounds, their name and CID were misaligned in table 1. This is corrected. For the first compound there is no assigned CID number; in all cases additional information is added to facilitate its identification.

-Table 1 – please explain See selected compounds (it has been written several times in the Table 1 ., it would be useful to explain in the Tables ́ title what it means, or put an asterisk and explain it under the table)

The meaning for the expression “selected compound” is explicited in the legend. -Table 1 – please include traditional use for Phyla strigulosa
Done

-You have stated that the initial criterion for further experiments (results showed in Table 2) was the extracts' ability to reduce the survival rate of MCF7 cell lines to less than 40% (so the rate of cell survival was from 0-40%) at a concentration of 50 μg/ml. How can you than have IC50=79.82 μg/mL for Piper ecuadorense Sodiro?

Local communities of Southern Ecuador widely use Piper ecuadorense Sodiro. It is for this reason that the alcoholic extract, even less toxic, was included in the list table 2. A sentence is added in the text to explain its selection:

-Please check the cited literature as it is not always listed in the same citation style.

Corrected
